# Prevalence and associated factors of last dental visit and teeth cleaning frequency in Bangladesh, Bhutan, and Nepal: Findings from nationally representative surveys

Rajat Das Gupta[1,2,3]*, Shams Shabab Haider[4], Shah Saif Jahan[2,5], Md. Irteja Islam[2,6,7,8], Ananna Mazumder[2,9], Muhammad Sohail Zafar[10,11,12,13], Nazeeba Siddika[2,14,15], Ehsanul Hoque Apu[2,15,16]

1 BRAC James P Grant School of Public Health, BRAC University, Dhaka, Bangladesh, 2 Centre for International Public Health and Environmental Research, Bangladesh (CIPHER,B), Dhaka, Bangladesh, 3 Department of Epidemiology and Biostatistics, Arnold School of Public Health, University of South Carolina, Columbia, South Carolina, United States of America, 4 Health Data Management Project, Friendship NGO, Dhaka, Bangladesh, 5 School of Allied Health, Faculty of Health, Education, Medicine and Social Care, Anglia Ruskin University, Chelmsford, United Kingdom, 6 Research, Innovation and Grants, Spreeha Bangladesh, Dhaka, Bangladesh, 7 Sydney School of Public Health, Faculty of Medicine and Health, The University of Sydney, Sydney, New South Wales, Australia, 8 Centre for Health Research and Faculty of Health, Engineering and Sciences, The University of Southern Queensland, Toowoomba, Queensland, Australia, 9 Jahurul Islam Medical College, Bajitpur, Kishoreganj, Bangladesh, 10 Department of Restorative Dentistry, College of Dentistry, Taibah University, Al Madinah Al Munawwarah, Saudi Arabia, 11 Centre of Medical and Bio-Allied Health Sciences Research, Ajman University, Ajman, United Arab Emirates, 12 School of Dentistry, University of Jordan, Amman, Jordan, 13 Department of Dental Materials, Islamic International Dental College, Riphah International University, Islamabad, Pakistan, 14 Department of Epidemiology and Biostatistics, College of Human Medicine, Michigan State University, East Lansing, Michigan, United States of America, 15 College of Dental Medicine, Lincoln Memorial University, Knoxville, Tennessee, United States of America, 16 Department of Biomedical Engineering, Institute of Quantitative Health Science and Engineering, Michigan State University, East Lansing, Michigan, United States of America

* rajat89.dasgupta@gmail.com

**Data Availability Statement:** The dataset of WHO STEPS is available in the WHO NCD microdata repository. The data can be accessed from the

## Abstract

This study evaluated the prevalence and frequency of teeth cleaning and last dental visits, along with associated socio-demographic factors, among residents of Bangladesh, Bhutan, and Nepal. The World Health Organization (WHO) STEPwise approach to surveillance (STEPS) survey data from Bangladesh (N = 8,164), Bhutan (N = 5,374) and Nepal (N = 5,371) were analyzed. After descriptive analysis, a multivariable multilevel logistic regression was conducted to identify the factors associated with oral hygiene. The following variables were considered as outcomes of interest: daily teeth cleaning frequency and visitation to the oral healthcare providers in the last six/twelve months at the time of data collection. Bangladesh had the highest proportion of respondents who cleaned their teeth at least once or twice a day, while Nepal had the lowest. Bhutan had the highest proportion of respondents who visited a dentist within the last six months (10.5%) or 12 months (16.0%). Almost 94.5% of Nepalese participants never visited a dentist. Participants of older age, who were females and had increased educational attainment, were more likely to follow oral hygiene measures. The populations of all the countries included in the study show poor adherence

following URL: https://extranet.who.int/ ncdsmicrodata/index.php/home (accessed on 30 April 2022). Following the instruction of the WHO NCD microdata repository, the data can be downloaded.

**Funding:** The authors received no specific funding for this work.

**Competing interests:** The authors have declared that no competing interests exist.

to oral hygiene practices. Health promotion programs should raise awareness regarding the advantages of regular teeth cleaning and dental check-ups.

## Introduction

Oral diseases have become a significant public health issue [1]. Since 1990, the global prevalence of oral diseases has increased by an average of 45.6%, disproportionately affecting lower socio-economic groups [1–3]. Dental caries, periodontal disease, and oropharyngeal cancers contributing significantly to the disease burden, affect approximately 3.9 billion people worldwide [4–6]. Moreover, poor oral health negatively impacts daily activities, work and educational performance, and quality of life, causing embarrassment, discomfort, pain, or dysfunction [7, 8]. Evidence suggests that oral diseases, especially periodontitis, are associated with systemic conditions, including cardiovascular disease, diabetes mellitus, and adverse pregnancy outcomes such as low birth weight [9–12]. Risk factors for oral diseases include poor oral hygiene, smoking and smokeless tobacco use, and an unhealthy diet [13]. Good oral hygiene practices, such as brushing teeth twice daily, flossing, and regular dental visits, significantly reduce the risk of oral diseases [14]. Additionally, adherence to these practices has been linked to higher socioeconomic status [15] and education level [8, 16].

Oral health is one of the most neglected areas of healthcare worldwide. In South Asia, an estimated 45% of people aged 35 to 44 years suffer from periodontitis alone [6]. There is a high prevalence of dental caries in Bangladesh, Bhutan, and Nepal [17–19]. Data from the World Health Organization (WHO) indicates that in 2022, the rates of untreated caries in permanent teeth for individuals aged 5 and older were 30.4% in Bangladesh, 29.7% in Bhutan, and 31.5% in Nepal [20–22]. Concurrently, the occurrence of severe periodontal disease among those aged 15 and above was recorded at 23.4% in Bangladesh, 14.8% in Bhutan, and 23.5% in Nepal during the same year [20–22]. Previous research indicates that many people in South Asian countries have inadequate oral health knowledge and follow outdated/traditional oral hygiene methods due to cultural, religious norms, and beliefs [3, 6, 17]. While dental brushing is widely practiced, flossing is not, and the use of fingers, chewing sticks, soot, and tobacco powder remains common. Moreover, regular dental checkups and tooth preservation are not prominent components of dental health behavior in South Asia, as many do not consider oral health to be vital [3, 23].

Previous research has consistently demonstrated a strong association between socioeconomic status (SES) and education to oral hygiene practices. Studies have shown that individuals with higher SES and educational attainment are more likely to engage in regular oral hygiene behaviors, such as brushing and flossing, and are more likely to visit dental care professionals regularly. Conversely, those with lower SES and less education often have poorer oral hygiene practices and are at greater risk for dental diseases. These disparities can be attributed to differences in access to dental care, health literacy, and overall health behaviors influenced by socioeconomic and educational factors [24–28].

Despite the extensive research linking sociodemographic factors to oral hygiene practices globally, few studies on oral hygiene practices and related sociodemographic factors are available for residents of Bangladesh, Nepal, and Bhutan [29, 30]. Moreover, these studies have not utilized nationally representative data. These countries present unique contextual variations that may influence the relationship between socioeconomic factors and oral hygiene behaviors. Cultural, economic, and healthcare system differences in these regions could result in distinct

patterns and determinants of oral health practices not fully captured by studies conducted in other contexts. Therefore, it is crucial to investigate these relationships within these specific countries to develop targeted and effective public health interventions. This study aims to investigate the prevalence of and factors (including the socio-demographic factors) associated with teeth cleaning frequency and dental visits in the past year among residents of these three South Asian countries (Bangladesh, Bhutan and Nepal), using nationally representative data.

## Materials and methods

### Study design, setting and sampling

A secondary analysis was conducted on the WHO STEPwise approach to surveillance (STEPS) surveys from Bangladesh, Bhutan, and Nepal. These nationally representative STEPS surveys aimed to update health indicators for selected non-communicable diseases in the respective countries. The most recent waves of the STEPS survey used were Bangladesh 2017–18, Bhutan 2019, and Nepal 2019 [18, 31, 32]. The surveys' methods, sampling strategy, sample size calculation, data collection procedure, and preliminary results have been published elsewhere [18, 31, 32].

A multistage stratified cluster sampling of households was employed in each country. The population was divided into strata based on urban and rural areas. Random selections of primary sampling units (PSUs) were made from both urban and rural areas. In all three countries, the PSUs were selected based on probability proportional to size methods. The sample size within each stratum was proportionate to the size of that subgroup in the national population, ensuring a representative sample. Participants were chosen from these selected households [18, 31, 32].

In Bhutan, 88 PSUs were selected, with 33 from urban and 55 from rural areas. In Bangladesh, 248 PSUs were selected from both urban and rural areas (496 PSUs in total). In Nepal, 259 PSUs were chosen, though the urban-rural proportion was not specified [18, 31, 32].

### Ethical considerations

The STEPS protocol was approved by a country-specific institutional review board in each country (Bangladesh: Bangladesh Medical Research Council; Bhutan: Research Ethics Board for Health; Nepal: Nepal Health Research Council). Written informed consent was obtained from participants before data collection. On the 8th April 2022, permission to use the dataset for this research project was granted by the NCD Microdata Repository of the WHO. The WHO team provided deidentified datasets for analysis to the researchers.

### Data collection

Data collection across three countries employed the WHO NCD STEPS instrument, incorporating three steps with core, expanded, and country-specific questions to measure NCD risk factors. In Bangladesh, the survey included essential and selected optional modules such as oral health and cervical cancer screening, and was translated into the local language with subsequent validation through back translation [18, 31, 32]. Trained enumerators and supervisors, who had undergone extensive training on interviewing techniques, household selection, and data recording on Android tablets, also received practical training on physical and biochemical measurements. Mock interviews were conducted in local dialects to ensure the quality and standardization of the process. Supervisors specifically focused on overseeing the data collection, ensuring data accuracy, and providing necessary support [18, 31, 32].

During the survey, participants were interviewed face-to-face to collect sociodemographic information, behavioral risk factors like smoking and alcohol use, and oral hygiene practices. Upon giving written consent, each participant completed STEP I and II questionnaires received a QR-coded urine container and a feedback form outlining their health metrics. Additionally, participants were given an appointment card detailing fasting instructions for subsequent biochemical measurements involving blood glucose and lipid assessments. However, this study did not include any biochemical measurements. The entire data collection process was facilitated using Android tablets with ODK software, and data were uploaded to a cloud-based server for analysis [18, 31, 32].

## Outcome variable

This study included two primary preventive oral health outcomes (i.e., teeth cleaning and dental visits) as they were available across the three surveys. The following variables were considered as outcomes of interest:

a. Cleaning teeth at least once a day

b. Cleaning teeth at least twice a day

c. Visited a dentist within the last 6 months

d. Visited a dentist within the last 12 months

e. Never visited a dentist

Each of these outcome variables was dichotomized into yes/no. The respondents were asked the following questions: (1) "How often do you clean your teeth?"; (2) "How long has it been since you last visited a dentist?".

"How often do you clean your teeth?" The respondents had the following options to answer:

I. Never

II. Once a month

III. 2–3 times a month

IV. Once a week

V. 2–6 times a week

VI. Once a day

VII. Twice or more a day

For cleaning teeth at least once a day 'Once a day' and 'Twice or more a day' were categorized as Yes = 1 and the rest as No = 0.

For cleaning teeth at least twice a day 'Twice or more a day' was categorized as Yes = 1 and the rest were categorized as No = 0.

For the question, "How long has it been since you last saw a dentist?", the respondents had the following options to answer:

I. Less than 6 months

II. 6–12 months

III. More than 1 year but less than 2 years

IV.  2 or more years but less than 5 years

V.  5 or more years

VI.  Never received dental care

For "never visiting a dentist", 'Never received dental care', was categorized as Yes = 1, and the rest were categorized as No = 0.

For "visiting a dentist in last twelve months", 'Less than 6 months' and '6–12 months' were categorized as Yes = 1, and the rest were categorized as No = 0.

For "visiting a dentist in last six months", 'Less than 6 months' was categorized as Yes = 1, and the rest were categorized as No = 0.

The categorization of the variables is data-driven as this is a secondary data analysis. Brushing one's teeth once daily is deemed adequate for preserving oral health and preventing dental caries and periodontal diseases. Additionally, tooth brushing plays a crucial role in applying anti-caries agents, such as fluorides. Nevertheless, many patients struggle to remove plaque through their at-home oral hygiene practices effectively. Consequently, most dentists/dental organizations, including the American Dental Association recommend brushing teeth twice daily to enhance plaque control [33, 34]. As such, we categorized the two outcome variables related to cleaning teeth into "cleaning teeth at least once a day" and "cleaning teeth at least twice a day". Most guidelines recommend a routine check-up by a dentist every six/twelve months [35, 36]. As such, we categorized the three outcome variables related to the last dental visit into visiting a dentist within the previous six months, within the last 12 months, and never visiting a dentist.

It is essential to highlight that two preventive oral health outcomes were considered across all three surveys: the frequency of teeth cleaning and dental visits. Both outcomes were included in our analysis. Additionally, descriptive statistics were provided for the primary reason individuals sought dental care, which was also available in the datasets of Bangladesh and Nepal. However, it should be noted that no other commonly measured oral health outcomes were present in the datasets from the three countries.

## Independent variables

The analysis included all the available sociodemographic factors in the STEPs datasets from the three countries. The following variables were considered as independent variables/confounders in every model: age group in years (18–29 years, 30–49 years, 50–69 years); gender (male, female); highest educational attainment (no formal schooling, up to primary, up to secondary, college and higher); marital status (never married, currently married, divorced/widowed/separated); smoking status (never smoker, current smoker, former smoker); and ever alcohol consumption (yes, no). For cleaning teeth at least once and at least twice a day, we included 'previous dental visit' as a covariate, which was categorized as follows: less than 6 months, 6–12 months, more than 12 months, and never visited (which included more than 1 year but less than 2 years, 2 or more years but less than 5 years; 5 or more years). For visiting a dentist within the last 6 or 12 months or never visiting a dentist, we used teeth cleaning frequency as a covariate, which was categorized into once a day, twice a day, and infrequent/never (which included never; once a month; 2–3 times a month; once a week; 2–6 times a week). Smoking within the 30 days before data collection was defined as current smoking [18, 31, 32]. The relationship between the variables is presented in a directed acyclic graph (DAG) in Fig 1. The DAG shows a bidirectional relationship between the last dental visit and the frequency of teeth cleaning. The paths that are marked as red imply confounding paths, which needed to be adjusted during the analysis.

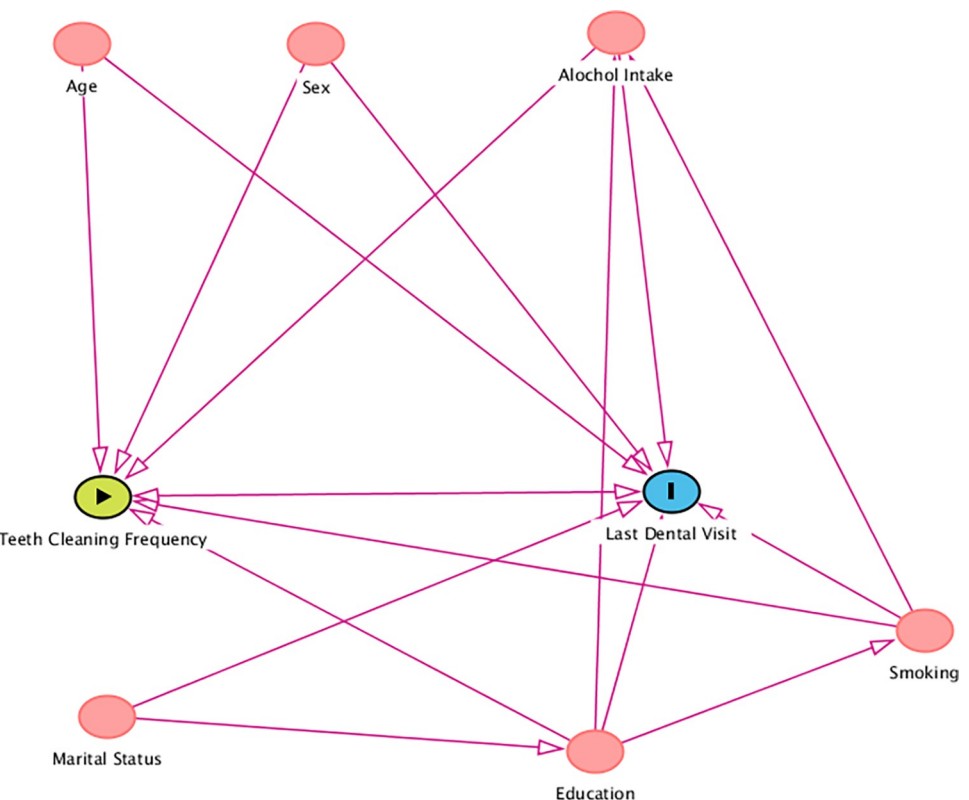

**Fig 1. A directed acyclic graph showing the relationship between the variables under study.**

## Statistical analysis

Complete case analysis was conducted given the very low percentage of missing data (<1%) [37]. At the beginning of the analysis, descriptive analyses were conducted. The results were presented in mean with standard deviation in the case of continuous variables and frequency with percentages in the case of categorical variables. During the analyses, adjusting for the cluster and strata-specific effect was achieved by incorporating the sampling weight provided by WHO STEPS. Then, bivariate analyses were conducted according to the different categories of the independent variables. Rao-Scott Chi-square tests were conducted to find the differences among the subcategories. Bivariate and multivariable multilevel logistic regressions were performed to identify the factors associated with oral hygiene practices. These analyses included adjustments for sociodemographic factors such as age, gender, and education to account for potential confounding variables. By adjusting for these factors, we aimed to ensure the robustness and reliability of our results, providing a more accurate understanding of the associations between the independent variables and oral hygiene practices. We reported both the crude odds ratio (COR) and the adjusted odds ratio (AOR), along with a 95% confidence interval (CI). We also conducted a log-binomial model and reported crude prevalence ratio (CPR) for the unadjusted model and adjusted prevalence ratio (APR) for the multivariable log-binomial model. 95% CI was reported for both CPR and APR. Stata version 17.0 was used for the data analyses (College Station, TX, USA).

## Findings

### Background characteristics of the participants

The background characteristics of the participants are presented in Table 1. In total, 18,909 participants were included in the final analysis (Bangladesh: N = 8,164; Bhutan: N = 5,374; and Nepal: N = 5,371). In all three countries, within each independent variable, most participants were aged between 18 and 49 years, were currently married, and had never been smokers. The proportion of males and females was almost the same in Bangladesh. Bhutan had a higher proportion of males, while Nepal had a higher proportion of females. Bhutan had the highest proportion of participants educated up to or above the secondary level (43.34%). Bangladesh had the highest percentage of current smokers (23.50%), followed by Nepal (18.26%), and Bhutan

**Table 1. Background characteristics of the participants (N = 18909).**

| Characteristics | Bangladesh (N = 8164) | | Bhutan (N = 5374) | | Nepal (N = 5371) | |
|---|---|---|---|---|---|---|
| | N | % | N | % | N | % |
| **Age Group (in years)** | | | | | | |
| 18–29 | 1941 | 42.40 | 1235 | 37.61 | 1245 | 40.48 |
| 30–49 | 4421 | 35.80 | 2765 | 44.25 | 2566 | 38.43 |
| 50–69 | 1802 | 21.80 | 1374 | 18.14 | 1560 | 21.09 |
| **Gender** | | | | | | |
| Male | 3804 | 49.33 | 2028 | 53.74 | 1910 | 46.50 |
| Female | 4360 | 50.67 | 3346 | 46.26 | 3461 | 53.50 |
| **Highest Educational Attainment** | | | | | | |
| No formal education | 2476 | 29.61 | 2823 | 40.64 | 2256 | 32.77 |
| Up to primary | 3735 | 45.31 | 798 | 16.02 | 1506 | 29.18 |
| Up to secondary | 1397 | 19.62 | 1362 | 35.15 | 1434 | 34.49 |
| College and higher | 556 | 5.46 | 391 | 8.19 | 175 | 3.55 |
| **Marital Status** | | | | | | |
| Never married | 534 | 12.33 | 595 | 20.65 | 337 | 13.49 |
| Currently married | 7237 | 82.67 | 4260 | 74.01 | 4732 | 83.60 |
| Divorced/widowed/separated | 393 | 5.00 | 519 | 5.33 | 302 | 2.91 |
| **Smoking Status** | | | | | | |
| Never smoker | 5619 | 69.54 | 4118 | 70.18 | 3–892 | 74.93 |
| Current smoker | 1922 | 23.50 | 354 | 10.58 | 1057 | 18.26 |
| Former smoker | 623 | 6.96 | 902 | 19.24 | 422 | 6.82 |
| **Ever Alcohol Consumption** | | | | | | |
| Yes | 626 | 8.64 | 3257 | 60.41 | 1561 | 29.63 |
| No | 7538 | 91.36 | 2117 | 39.59 | 3810 | 70.37 |
| **Last Dental Visit** | | | | | | |
| Less than 6 months | 668 | 7.27 | 555 | 10.46 | 126 | 1.52 |
| 6–12 months | 473 | 5.93 | 284 | 5.53 | 91 | 1.33 |
| More than 12 months | 1395 | 15.88 | 1810 | 32.84 | 229 | 2.71 |
| Never Visited | 5628 | 70.93 | 2725 | 51.16 | 4925 | 94.45 |
| **Teeth Cleaning Frequency** | | | | | | |
| Once a day | 4006 | 53.07 | 1841 | 67.94 | 4319 | 82.71 |
| Twice a day | 4098 | 46.12 | 617 | 23.08 | 359 | 6.63 |
| Infrequent | 60 | 0.81 | 295 | 8.98 | 693 | 10.66 |

The frequencies are unweighted and the proportions are weighted

(10.58%). Bhutan had the highest (60.41%) and Bangladesh had the lowest (8.64%) proportions of participants who had ever consumed alcohol.

## Teeth cleaning frequency

Table 2 presents the proportions of respondents regarding cleaning their teeth at least once or twice a day. Bangladesh had the highest proportion of respondents who cleaned their teeth at least once (99.19%) or twice (46.12%) a day, while Nepal had the lowest 89.34% and 6.63%,

**Table 2. Distribution of the respondents regarding cleaning teeth at least once a day or twice a day.**

| Characteristics | Cleaning teeth at least once a day; % (95% CI) | | | Cleaning teeth at least twice a day; % (95% CI) | | |
|---|---|---|---|---|---|---|
| | Bangladesh | Bhutan | Nepal | Bangladesh | Bhutan | Nepal |
| **Total** | 99.19 (98.81–99.45) | 91.02 (88.89–92.77) | 89.34 (86.88–91.34) | 46.12 (43.68–48.58) | 23.08 (20.34–26.06) | 6.63 (5.11–8.55) |
| **Age Group (in years)** | | | | | | |
| 18–29 | 99.65 (99.13–99.86) | 96.51 (93.70–98.09) | 95.34 (92.65–97.08) | 42.65 (39.04–46.34) | 28.32 (23.35–33.87) | 9.54 (7.06–12.78) |
| 30–49 | 99.34 (98.88–99.62) | 92.78 (90.56–94.51) | 88.74 (85.16–91.54) | 47.99 (45.48–50.51) | 22.18 (19.45–25.17) | 5.27 (3.67–7.51) |
| 50–69 | 98.03 (96.87–98.77) | 74.33 (68.76–79.21) | 78.92 (74.40–82.83) | 49.81 (45.48–54.14) | 14.32 (10.72–18.89) | 3.51 (2.37–5.17) |
| *p*-value | <0.0001 | <0.0001 | <0.0001 | 0.0041 | 0.0001 | <0.0001 |
| **Gender** | | | | | | |
| Male | 98.61 (97.86–99.10) | 89.60 (86.74–91.89) | 89.44 (86.24–91.97) | 36.92 (33.95–40.00) | 19.82 (16.77–23.27) | 5.78 (4.00–8.29) |
| Female | 99.75 (99.45–99.89) | 92.77 (90.63–94.45) | 89.25 (86.66–91.39) | 55.08 (51.93–58.19) | 27.10 (23.54–30.99) | 7.36 (5.55–9.71) |
| *p*-value | <0.0001 | 0.0105 | 0.8868 | <0.0001 | 0.0005 | 0.2063 |
| **Highest Educational Attainment** | | | | | | |
| No Formal Education | 98.31 (97.45–98.89) | 84.78 (81.28–87.73) | 82.44 (78.37–85.88) | 46.17 (42.02–50.38) | 14.88 (12.08–18.20) | 3.70 (2.41–5.65) |
| Up to primary | 99.47 (99.04–99.71) | 91.28 (86.60–94.43) | 90.61 (86.89–93.36) | 43.93 (41.02–46.89) | 20.81 (16.11–26.45) | 4.67 (3.28–6.61) |
| Up to secondary | 99.62 (98.87–99.88) | 99.29 (98.38–99.69) | 93.92 (90.86–96.00) | 47.42 (42.82–52.06) | 34.05 (29.07–39.40) | 9.67 (6.99–13.24) |
| College and higher | 100 (-) | 98.20 (92.75–99.57) | 98.10 (94.10–99.41) | 59.37 (51.75–66.56) | 40.63 (32.60–49.20) | 20.14 (11.89–32.03) |
| *p*-value | <0.0001 | <0.0001 | <0.0001 | 0.0043 | <0.0001 | <0.0001 |
| **Marital Status** | | | | | | |
| Never married | 99.68 (97.75–99.95) | 89.98 (87.75–91.84) | 95.62 (91.62–97.75) | 39.13 (33.31–45.27) | 21.73 (18.87–24.89) | 12.14 (7.65–18.74) |
| Currently married | 99.07 (98.62–99.37) | 97.12 (94.18–98.60) | 88.85 (86.22–91.03) | 46.17 (43.63–48.72) | 26.62 (18.14–37.26) | 5.85 (4.55–7.49) |
| Divorced/widowed/separated | 99.98 (99.82–100.00) | 91.43 (87.62–94.14) | 74.35 (65.02–81.89) | 62.67 (53.08–71.35) | 26.27 (20.36–33.18) | 3.39 (1.74–6.51) |
| *p*-value | 0.1296 | 0.0006 | <0.0001 | 0.0002 | 0.3004 | 0.0003 |
| **Smoking Status** | | | | | | |
| Never Smoker | 99.81 (.99.61-.99.91) | 96.90 (93.91–98.45) | 91.47 (89.11–93.36) | 49.58 (46.85–52.30) | 35.64 (28.23–43.80) | 7.38 (5.68–9.55) |
| Current Smoker | 97.91 (.96.69-.98.68) | 89.97 (87.52–91.98) | 82.42 (77.58–86.41) | 37.13 (33.18–41.26) | 20.52 (18.02–23.26) | 4.24 (2.46–7.21) |
| Former Smoker | 97.33 (.94.17-.98.8) | 85.94 (80.52–90.03) | 84.43 (77.80–89.36) | 41.99 (36.22–47.99) | 17.05 (12.18–23.36) | 4.70 (2.76–7.90) |
| *p*-value | <0.0001 | 0.0001 | <0.0001 | <0.0001 | <0.0001 | 0.0172 |
| **Ever Alcohol Consumption** | | | | | | |
| Yes | 98.26 (94.48–99.47) | 89.26 (85.92–91.88) | 86.29 (82.27–89.50) | 33.11 (27.58–39.14) | 19.97 (16.74–23.65) | 7.03 (4.81–10.15) |
| No | 99.28 (98.97–99.49) | 93.53 (91.24–95.26) | 90.63 (87.99–92.73) | 47.35 (44.84–49.88) | 27.53 (23.34–32.16) | 6.46 (4.86–8.54) |
| *p*-value | 0.1348 | 0.0179 | 0.0119 | <0.0001 | 0.0063 | 0.6708 |
| **Last Dental Visit** | | | | | | |
| Less than 6 months | 99.47 (98.57–99.81) | 93.23 (68.48–98.87) | 84.88 (73.67–91.85) | 44.50 (39.08–50.06) | 47.71 (10.18–88.01) | 9.86 (4.18–21.53) |
| 6–12 months | 99.88 (99.13–99.98) | 100.00 N/A | 86.91 (68.88–95.22) | 46.12 (38.49–53.95) | 63.18 (29.33–87.65) | 7.96 (3.09–19.03) |
| More than 12 months | 98.63 (97.54–99.25) | 79.29 (47.30–94.23) | 86.99 (79.40–92.06) | 48.77 (44.39–53.18) | 28.64 (6.16–71.05) | 6.06 (2.03–16.72) |
| Never Visited | 99.23 (98.7–99.54) | 91.02 (88.87–92.79) | 89.51 (86.97–91.61) | 45.70 (42.85–48.57) | 22.87 (20.15–25.83) | 6.57 (5.03–8.54) |
| *p*-value | 0.0907 | 0.5565 | 0.5866 | 0.5882 | 0.0926 | 0.8034 |

CI: Confidence Interval

respectively. Approximately half of the Bangladeshi participants (46.12%; 95% CI: 43.68%-48.58%) cleaned their teeth twice daily, compared to only 6.63% (95% CI: 5.11%-8.55%) of Nepalese participants. The prevalence of teeth cleaning at least once or twice a day significantly increased with decreasing age, as well as increasing educational attainment. Never smokers had a significantly higher proportion of members who cleaned their teeth at least once or twice a day. Although females had a higher prevalence of teeth cleaning in all three countries than their male counterparts, significant differences were only found in Bangladesh and Bhutan. The prevalence differed for different marital status categories in other countries. In Bangladesh and Nepal, most individuals use toothpaste and toothbrushes to clean their teeth. However, in Bangladesh, approximately 22.16% (95% CI: 19.98–24.50) of respondents also used charcoal to clean their teeth (S1 Table).

The proportions of the respondents regarding visiting a dentist within the last 6 or 12 months or never visiting a dentist are shown in Table 3. Bhutan has the highest proportion of respondents who visited a dentist within the previous 6 months (10.46%; 95% CI: 9.15%-11.94%) or within the last 12 months (15.99%; 95% CI: 14.26%-17.90%). Almost 94.45% of Nepalese participants (95% CI: 93.17%-95.50%) never visited a dentist. The proportion varied among the covariates. In Bangladesh/Nepal, the main reason for visiting a dentist was 'pain or trouble with teeth, gums, or the mouth'. Only 1.07% of Bangladeshi adults and 2.51% of Nepalese adults visited the dentist for a routine checkup (S1 and S2 Tables).

**Factors associated with teeth cleaning.** The results of multivariable logistic regression analyses for the aspects related to cleaning teeth at least once and twice a day in Bangladesh, Bhutan, and Nepal are shown in Table 4. The detailed estimates from the logistic regression and log-binomial models are shown in S3–S8 Tables. Although there was a variation in the associated factors among the countries, all the covariates were associated with teeth cleaning.

**Cleaning teeth at least once a day.** The odds of teeth cleaning at least once a day decreased with age. Nepalese participants aged 30–49 years were 50% less likely to wash their teeth at least once daily (AOR: 0.50; 95% CI: 0.35–0.72, p<0.001) compared to the age group 18–29 years. The participants of the age group 50–69 years from Bhutan and Nepal had approximately 80% fewer odds compared to 18–29 years old (Bhutan: AOR: 0.21; 95% CI: 0.12–0.37, p<0.001; Nepal: AOR: 0.23; 95% CI: 0.15–0.34, p<0.001). In Bangladesh and Bhutan, females clean their teeth at least once a day more often than males; a significant association was only found in Bhutan (AOR: 1.69; 95% CI: 1.27–2.25, p<0.001). The odds of teeth cleaning at least once a day increased with increasing educational attainment.

Currently married individuals in Bhutan were twice as likely to clean their teeth at least once daily (AOR: 2.08; 95% CI: 1.14–3.80, p<0.001) than the individuals who never got married. In both Bangladesh and Nepal, current smokers (Bangladesh: AOR: 0.22; 95% CI: 0.08–0.60, p<0.001; Nepal: AOR: 0.49; 95% CI: 0.38–0.63, p<0.001) and former smokers (Bangladesh: AOR: 0.20; 95% CI: 0.06–0.61, p<0.001; Nepal: AOR: 0.66; 95% CI: 0.47–0.94, p<0.001) were less likely to clean their teeth at least once a day compared to never smokers. In Bhutan and Nepal, those who never consumed alcohol were more likely to clean their teeth once a day than those who had ever consumed alcohol (Bhutan: AOR: 1.6; 95% CI:1.2–2.2, p<0.001; Nepal: AOR: 1.5; 95% CI: 1.2–2.0, p<0.001). The last dental visit was not significantly associated with cleaning teeth at least once a day.

**Cleaning teeth at least twice a day.** In Bangladesh, the odds of cleaning one's teeth at least twice a day increased with increasing age (30–49 years: AOR: 1.42; 95% CI: 1.24–1.62, p<0.001; 50–69 years: AOR: 1.56; 95% CI: 1.32–1.85, p<0.001) compared to 18-29-year-old individuals. In each country, the odds were significantly higher among females (compared to males) and higher educational categories (compared to no formal education). In Bhutan, the odds of teeth cleaning at least twice a day were 37% higher among former smokers (AOR: 1.37;

**Table 3. Distribution of the respondents regarding visiting a dentist within the last 6 or 12 months or never visiting a dentist.**

| Characteristics | Visited a dentist within the last 6 months; % (95% CI) | | | Visited a dentist within the last 12 months; % (95% CI) | | | Never visited a Dentist; % (95% CI) | | |
|---|---|---|---|---|---|---|---|---|---|
| | Bangladesh | Bhutan | Nepal | Bangladesh | Bhutan | Nepal | Bangladesh | Bhutan | Nepal |
| **Total** | 7.27 (6.41–8.23) | 10.46 (9.15–11.94) | 1.52 (1.11–2.06) | 13.19 (12.07–14.40) | 15.99 (14.26–17.90) | 2.84 (2.13–3.79) | 70.93 (69.21–72.58) | 51.16 (48.21–54.10) | 94.45 (93.17–95.50) |
| **Age Group (in years)** | | | | | | | | | |
| 18–29 | 6.04 (4.71–7.72) | 12.62 (10.05–15.72) | 0.80 (0.44–1.42) | 11.30 (9.42–13.50) | 18.36 (15.40–21.74) | 1.79 (1.11–2.88) | 77.81 (75.16–80.24) | 49.22 (44.90–53.56) | 96.96 (95.59–97.91) |
| 30–49 | 7.99 (6.97–9.14) | 9.34 (7.85–11.07) | 1.95 (1.31–2.88) | 14.17 (12.80–15.65) | 14.72 (12.85–16.80) | 3.20 (2.30–4.44) | 67.30 (64.95–69.57) | 54.17 (50.44–57.85) | 93.51 (91.78–94.91) |
| 50–69 | 8.46 (6.77–10.53) | 8.74 (6.91–10.99) | 2.11 (1.39–3.20) | 15.27 (12.64–18.35) | 14.21 (11.93–16.86) | 4.21 (2.76–6.37) | 63.50 (59.49–67.33) | 47.85 (43.86–51.87) | 91.34 (88.55–93.50) |
| p-value | 0.042 | 0.0229 | 0.0041 | 0.0283 | 0.0191 | 0.0045 | <0.0001 | 0.0223 | <0.0001 |
| **Gender** | | | | | | | | | |
| Male | 7.35 (6.26–8.61) | 8.87 (7.43–10.55) | 0.93 (0.54–1.61) | 12.81 (11.33–14.45) | 14.80 (12.67–17.23) | 1.61 (0.98–2.61) | 70.96 (68.57–73.24) | 52.55 (48.94–56.14) | 96.34 (94.91–97.38) |
| Female | 7.19 (6.04–8.53) | 12.31 (10.30–14.66) | 2.02 (1.45–2.82) | 13.57 (11.98–15.33) | 17.38 (15.20–19.80) | 3.92 (2.85–5.35) | 70.90 (68.71–72.99) | 49.54 (46.22–52.87) | 92.80 (91.03–94.25) |
| p-value | 0.8432 | 0.0063 | 0.0089 | 0.5068 | 0.0712 | 0.0006 | 0.9657 | 0.1126 | <0.0001 |
| **Highest Educational Attainment** | | | | | | | | | |
| No Formal Education | 7.03 (5.70–8.65) | 7.96 (6.53–9.68) | 1.83 (1.23–2.71) | 11.92 (10.10–14.01) | 12.51 (10.94–14.27) | 3.51 (2.47–4.96) | 72.90 (70.11–75.52) | 58.87 (55.76–61.92) | 92.48 (90.24–94.24) |
| Up to primary | 7.59 (6.36–9.02) | 7.20 (5.10–10.07) | 1.48 (0.85–2.57) | 14.36 (12.65–16.24) | 11.09 (8.43–14.46) | 2.30 (1.45–3.63) | 70.03 (67.63–72.33) | 56.28 (51.23–61.20) | 95.04 (93.20–96.40) |
| Up to secondary | 6.64 (4.77–9.18) | 12.96 (10.72–15.58) | 1.25 (0.76–2.06) | 11.58 (9.04–14.72) | 19.20 (16.52–22.21) | 2.72 (1.76–4.16) | 72.62 (68.61–76.29) | 44.06 (39.48–48.75) | 95.66 (93.89–96.93) |
| College and higher | 8.16 (5.37–12.21) | 18.52 (13.35–25.11) | 1.45 (0.38–5.33) | 16.29 (12.10–21.58) | 29.08 (22.12–37.19) | 2.39 (0.89–6.25) | 61.59 (54.32–68.37) | 33.34 (27.49–39.76) | 96.02 (91.66–98.15) |
| p-value | 0.7969 | <0.0001 | 0.6644 | 0.1071 | <0.0001 | 0.338 | 0.0137 | <0.0001 | <0.0001 |
| **Marital Status** | | | | | | | | | |
| Never married | 6.47 (4.27–9.69) | 10.12 (8.61–11.87) | 0.55 (0.20–1.54) | 12.03 (8.93–16.01) | 15.71 (13.71–17.95) | 1.26 (0.49–3.25) | 77.59 (72.37–82.07) | 51.86 (48.66–55.06) | 97.84 (95.72–98.93) |
| Currently married | 7.53 (6.61–8.55) | 9.59 (6.76–13.44) | 1.65 (1.20–2.27) | 13.38 (12.22–14.62) | 13.74 (10.23–18.22) | 3.09 (2.27–4.18) | 70.38 (68.61–72.09) | 53.25 (46.22–60.16) | 94.01 (92.53–95.21) |
| Divorced/widowed/separated | 4.96 (2.99–8.11) | 12.18 (9.11–16.11) | 2.05 (0.80–5.15) | 13.06 (8.02–20.55) | 18.27 (14.45–22.82) | 3.16 (1.53–6.42) | 63.59 (54.24–72.02) | 47.45 (42.52–52.43) | 91.48 (84.19–95.58) |
| p-value | 0.3109 | 0.4135 | 0.0558 | 0.796 | 0.2767 | 0.0894 | 0.0086 | 0.2041 | 0.0042 |
| **Smoking Status** | | | | | | | | | |
| Never Smoker | 6.87 (5.90–7.99) | 14.09 (10.54–18.59) | 1.18 (0.83–1.69) | 13.26 (11.87–14.79) | 19.32 (15.14–24.32) | 2.32 (1.67–3.20) | 70.91 (68.83–72.91) | 45.01 (38.78–51.41) | 95.42 (94.22–96.38) |
| Current Smoker | 7.94 (6.24–10.06) | 9.54 (8.40–10.82) | 2.23 (1.18–4.17) | 12.15 (10.01–14.66) | 15.27 (13.56–17.16) | 4.08 (2.53–6.52) | 72.67 (69.47–75.65) | 52.95 (49.83–56.05) | 92.92 (90.35–94.85) |
| Former Smoker | 8.92 (6.13–12.81) | 9.12 (6.50–12.67) | 3.25 (1.58–6.57) | 16.04 (12.20–20.80) | 13.16 (9.82–17.41) | 5.27 (2.93–9.30) | 65.20 (59.50–70.48) | 50.16 (43.94–56.38) | 87.90 (82.05–92.02) |
| p-value | 0.3245 | 0.0091 | 0.023 | 0.2887 | 0.0615 | 0.0075 | 0.0696 | 0.0215 | <0.0001 |
| **Ever Alcohol Consumption** | | | | | | | | | |
| Yes | 6.62 (4.62–9.40) | 10.13 (8.61–11.89) | 1.34 (0.71–2.51) | 11.66 (8.58–15.67) | 15.65 (13.58–17.96) | 2.91 (1.80–4.68) | 71.96 (65.66–77.49) | 49.94 (46.43–53.45) | 94.33 (91.92–96.05) |
| No | 7.33 (6.43–8.34) | 10.96 (9.01–13.26) | 1.59 (1.15–2.20) | 13.34 (12.14–14.63) | 16.52 (13.92–19.50) | 2.81 (2.12–3.72) | 70.83 (68.07–72.53) | 53.02 (49.34–56.66) | 94.50 (93.19–95.57) |
| p-value | 0.59 | 0.5042 | 0.6187 | 0.4046 | 0.6031 | 0.8775 | 0.7204 | 0.1527 | 0.8674 |

(*Continued*)

**Table 3.** (Continued)

| Characteristics | Visited a dentist within the last 6 months; % (95% CI) | | | Visited a dentist within the last 12 months; % (95% CI) | | | Never visited a Dentist; % (95% CI) | | |
|---|---|---|---|---|---|---|---|---|---|
| | Bangladesh | Bhutan | Nepal | Bangladesh | Bhutan | Nepal | Bangladesh | Bhutan | Nepal |
| **Teeth Cleaning Frequency** | | | | | | | | | |
| Once a day | 7.53 (6.38–8.87) | 0.86 (0.19–3.90) | 1.37 (0.96–1.97) | 13.53 (11.91–15.33) | 0.30 (0.08–1.08) | 2.64 (1.88–3.69) | 71.55 (68.94–74.02) | 99.42 (98.12–99.82) | 94.71 (93.32–95.82) |
| Twice a day | 7.01 (5.92–8.29) | 2.70 (3.70–18.70) | 2.25 (0.98–5.08) | 12.94 (11.47–14.56) | 1.32 (0.30–5.56) | 3.85 (2.08–7.02) | 70.27 (68.23–72.24) | 98.21 (91.85–99.63) | 93.67 (89.28–96.34) |
| Infrequent | 4.71 (1.77–11.91) | 0.97 (1.30–7.20) | 2.15 (1.12–4.09) | 5.60 (2.09–14.17) | 0.10 (0.01–0.72) | 3.78 (2.04–6.88) | 67.66 (49.79–81.53) | 99.03 (91.85–99.63) | 92.91 (89.04–95.49) |
| *p*-value | 0.565 | 0.2743 | 0.3018 | 0.2963 | 0.0179 | 0.352 | 0.6516 | 0.1187 | 0.4117 |

CI: Confidence Interval

95% CI: 1.04–1.80, p<0.001), compared to non-smokers and 48% higher among those who never consumed alcohol (AOR: 1.48; 95% CI: 1.20–1.82, p<0.001), compared to those who had ever consumed alcohol. The last dental visit was not significantly associated with cleaning teeth at least twice a day.

## Dental visits

The results of multivariable logistic regression analyses for the factors associated with visiting a dentist in the last six months, twelve months, or never visiting a dentist in Bangladesh, Bhutan, and Nepal are shown in Table 5. The detailed logistic regression and log-binomial models are shown in S9–S17 Tables. In Bangladesh and Nepal, the odds of visiting a dentist in the last six/twelve months were significantly higher among the participants aged 30–49 years and 50–69 years compared to the 18-29-year-old participants. The odds of never visiting a dentist decreased significantly with increasing age in all three nations.

Females had higher odds of visiting a dentist in the last six/twelve months and lower odds of never seeing a dentist than males, with a significant association found in Bhutan and Nepal. Females in Nepal had 50% (AOR: 0.50; 95% CI: 0.37–0.67, p<0.001) lower odds of never visiting a dentist than males, respectively.

The odds of visiting a dentist in the last six/twelve months increased with increasing educational attainment. An inverse direction of odds was found for never visiting a dentist. A significant association was found in the case of Bangladesh and Bhutan. Among the Nepalese participants, current smokers and former smokers had significantly higher odds of visiting a dentist compared to never smokers, and the odds of never visiting a dentist were 310% and 45% lower for the current smokers (AOR: 0.69; 95% CI: 0.50–0.93p<0.001) and the former smokers (AOR: 0.55; 95% CI: 0.38–0.80, p<0.001) respectively, compared to the never smokers. Frequency of teeth cleaning was not significantly associated with visiting a dentist in the last six months, or twelve months, or never visiting a dentist in any of the three countries.

The logistic regression model and log-binomial model provided findings that were almost similar. Most factors that exhibited statistical significance based on odds ratios were similarly found to be significant when assessed using prevalence ratios. However, the magnitude of the association was different as the magnitude of association was less in prevalence ratios with narrower CI compared to the odds ratio (S3–S17 Tables).

**Table 4. Results of multivariable logistic regression analyses to compare adjusted odds ratio for the factors associated with cleaning teeth at least once and twice a day in Bangladesh- Bhutan and Nepal.**

| Variables | Cleaning teeth at least once a day | | | Cleaning teeth at least twice a day | | |
|---|---|---|---|---|---|---|
| | Bangladesh | Bhutan | Nepal | Bangladesh | Bhutan | Nepal |
| | AOR (95% CI) | AOR (95% CI) | AOR (95% CI) | AOR (95% CI) | AOR (95% CI) | AOR (95% CI) |
| **Age Group (in years)** | | | | | | |
| 18–29 | Ref | Ref | Ref | Ref | Ref | Ref |
| 30–49 | 1.59 (0.64–3.92) | 0.61 (0.36–1.04) | 0.50*** (0.35–0.72) | 1.42*** (1.24–1.62) | 1.05 (0.81–1.36) | 0.74 (0.54–1.01) |
| 50–69 | 0.68 (0.27–1.73) | 0.21*** (0.12–0.37) | 0.23*** (0.15–0.34) | 1.56*** (1.32–1.85) | 0.77 (0.54–1.09) | 0.72 (0.47–1.11) |
| **Gender** | | | | | | |
| Male | Ref | Ref | Ref | Ref | Ref | Ref |
| Female | 1.44 (0.48–4.28) | 1.69*** (1.27–2.25) | 0.93 (0.72–1.20) | 2.56*** (2.22–2.96) | 1.66*** (1.33–2.08) | 1.65** (1.20–2.27) |
| **Highest Educational Attainment** | | | | | | |
| No Formal Education | Ref | Ref | Ref | Ref | Ref | Ref |
| Up to primary | 2.33** (1.25–4.35) | 1.81** (1.19–2.74) | 1.99*** (1.52–2.60) | 1.05 (0.92–1.18) | 1.64 (1.24–2.18) | 1.76 (1.21–2.58) |
| Up to secondary | 4.52** (1.48–13.77) | 7.67*** (3.72–15.80) | 3.40*** (2.35–4.92) | 1.47*** (1.25–1.73) | 3.00*** (2.30–3.91) | 3.10*** (2.11–4.55) |
| College and higher | 1.00 (-) | 6.12** (1.84–20.39) | 3.16* (1.32–7.57) | 2.48*** (1.97–3.13) | 4.41*** (2.90–6.72) | 7.10*** (4.04–12.45) |
| **Marital Status** | | | | | | |
| Never married | Ref | Ref | Ref | Ref | Ref | Ref |
| Currently married | 0.36 (0.04–3.03) | 2.08* (1.14–3.80) | 1.30 (0.71–2.37) | 0.94 (0.75–1.18) | 0.80 (0.57–1.12) | 0.85 (0.54–1.34) |
| Divorced/widowed/separated | 1.25 (0.06–27.52) | 1.81 (0.89–3.68) | 0.65 (0.32–1.30) | 1.01 (0.72–1.42) | 0.79 (0.49–1.27) | 0.91 (0.42–1.98) |
| **Smoking Status** | | | | | | |
| Never Smoker | Ref | Ref | Ref | Ref | Ref | Ref |
| Current Smoker | 0.22** (0.08–0.60) | 1.56 (0.70–3.48) | 0.49*** (0.38–0.63) | 0.93 (0.79–1.09) | 1.02 (0.69–1.52) | 0.70 (0.46–1.08) |
| Former Smoker | 0.20** (0.06–0.61) | 1.02 (0.70–1.48) | 0.66* (0.47–0.94) | 1.06 (0.85–1.31) | 1.37*** (1.04–1.80) | 1.34 (0.81–2.21) |
| **Ever Alcohol Consumption** | | | | | | |
| Yes | Ref | Ref | Ref | Ref | Ref | Ref |
| No | 0.73 (0.28–1.90) | 1.63** (1.20–2.21) | 1.52** (1.17–1.98) | 1.03 (0.84–1.26) | 1.48*** (1.20–1.82) | 0.96 (0.69–1.34) |
| **Dental Visit** | | | | | | |
| Less than 6 months | Ref | Ref | Ref | Ref | Ref | Ref |
| 6–12 months | 5.39 (0.58–50.37) | 1.00 (-) | 0.99 (0.37–2.66) | 0.85 (0.65–1.10) | 3.54 (0.20–63.06) | 1.72 (0.59–5.06) |
| More than 12 months | 1.18 (0.40–3.46) | 1.10 (0.09–13.92) | 1.18 (0.54–2.56) | 0.90 (0.73–1.11) | 0.70 (0.08–6.41) | 1.03 (0.41–2.59) |
| Never Visited | 1.69 (0.65–4.37) | 2.45 (0.25–24.13) | 1.37 (0.73–2.57) | 0.88 (0.73–1.05) | 0.87 (0.14–5.37) | 0.82 (0.40–1.69) |

AOR: Adjusted Odds Ratio; CI: Confidence Interval

*: p<0.05; **:p<0.01; ***:p<0.001

## Discussion

Using nationally representative samples from Bangladesh, Bhutan, and Nepal, we found that Bangladesh had the highest proportion of respondents who cleaned their teeth at least once (99.2%) or twice a day (46.1%); in contrast, Nepal had the lowest. Bhutan had the highest proportion of respondents who visited a dentist within the last 6 months (10.5% compared to Nepal's 1.5%) and the last 12 months (16.0% compared to Nepal's 2.8%). Meanwhile, approximately 95% of Nepalese adults never visited a dentist. We found that younger age, female gender, and increasing educational attainment were significantly associated with cleaning one's teeth at least once a day. In Bangladesh, older age was significantly associated with cleaning one's teeth at least twice a day. Participants aged 30–69 years (in Bangladesh and Nepal), of female gender (in Bhutan and Nepal), with increasing educational attainment (in Bangladesh

**Table 5. Results of multivariable logistic regression analyses to compare adjusted odds ratio for the factors associated with visiting a dentist in last six months-twelve months or never visiting a dentist in Bangladesh- Bhutan and Nepal.**

| Variables | Visited a dentist within the last 6 months | | | Visited a dentist within the last 12 months | | | Never visited a Dentist | | |
|---|---|---|---|---|---|---|---|---|---|
| | Bangladesh | Bhutan | Nepal | Bangladesh | Bhutan | Nepal | Bangladesh | Bhutan | Nepal |
| | AOR (95% CI) | AOR (95% CI) | AOR (95% CI) | AOR (95% CI) | AOR (95% CI) | AOR (95% CI) | AOR (95% CI) | AOR (95% CI) | AOR (95% CI) |
| **Age Group (in years)** | | | | | | | | | |
| 18–29 | Ref | Ref | Ref | Ref | Ref | Ref | Ref | Ref | Ref |
| 30–49 | 1.51** (1.19–1.91) | 0.86 (0.12–6.01) | 2.32* (1.22–4.43) | 1.47*** (1.23–1.77) | 0.51 (0.11–2.34) | 1.98** (1.23–3.19) | 0.55*** (0.48–0.63) | 0.73 (0.23–2.35) | 0.41*** (0.29–0.59) |
| 50–69 | 1.70*** (1.28–2.27) | 1.00 (-) | 2.72** (1.29–5.71) | 1.67*** (1.34–2.10) | 1.00 (-) | 2.54** (1.46–4.42) | 0.42*** (0.36–0.51) | 1.57 (0.33–7.41) | 0.32*** (0.21–0.48) |
| **Gender** | | | | | | | | | |
| Male | Ref | Ref | Ref | Ref | Ref | Ref | Ref | Ref | Ref |
| Female | 1.12 (0.88–1.42) | 1.19 (0.16–8.99) | 1.86* (1.14–3.04) | 1.16 (0.97–1.41) | 0.61 (0.13–2.74) | 1.93** (1.30–2.86) | 0.90 (0.78–1.04) | 2.03 (0.77–5.31) | 0.50*** (0.37–0.67) |
| **Highest Educational Attainment** | | | | | | | | | |
| No Formal Education | Ref | Ref | Ref | Ref | Ref | Ref | Ref | Ref | Ref |
| Up to primary | 1.38** (1.12–1.70) | 1.00 (-) | 1.33 (0.81–2.19) | 1.41*** (1.20–1.67) | 1.00 (-) | 0.96 (0.64–1.44) | 0.66*** (0.58–0.75) | 1.17 (0.24–5.66) | 0.84 (0.63–1.13) |
| Up to secondary | 1.42* (1.08–1.85) | 0.15 (0.00–5.62) | 1.42 (0.79–2.57) | 1.42** (1.15–1.76) | 0.55 (0.08–3.97) | 1.34 (0.84–2.12) | 0.58*** (0.49–0.68) | 1.56 (0.39–6.28) | 0.73 (0.52–1.03) |
| College and higher | 1.51* (1.06–2.14) | 1.82 (0.10–32.81) | 1.24 (0.34–4.54) | 1.78*** (1.36–2.33) | 0.88 (0.06–13.44) | 1.25 (0.50–3.11) | 0.38*** (0.31–0.48) | 0.54 (0.07–4.11) | 0.86 (0.44–1.69) |
| **Marital Status** | | | | | | | | | |
| Never married | Ref | Ref | Ref | Ref | Ref | Ref | Ref | Ref | Ref |
| Currently married | 0.85 (0.58–1.24) | 0.12 (0.00–5.57) | 0.94 (0.34–2.55) | 0.84 (0.62–1.14) | 0.26 (0.02–2.80) | 0.96 (0.45–2.06) | 0.99 (0.78–1.26) | 1.05 (0.20–5.50) | 1.02 (0.58–1.79) |
| Divorced/widowed/separated | 0.74 (0.42–1.30) | 0.18 (0.00–23.36) | 0.90 (0.26–3.19) | 0.73 (0.47–1.14) | 0.15 (0.00–6.92) | 0.61 (0.22–1.69) | 0.88 (0.63–1.25) | 3.40 (0.19–60.14) | 1.09 (0.53–2.24) |
| **Smoking Status** | | | | | | | | | |
| Never Smoker | Ref | Ref | Ref | Ref | Ref | Ref | Ref | Ref | Ref |
| Current Smoker | 0.97 (0.73–1.27) | 1.00 (-) | 1.66 (0.98–2.81) | 0.93 (0.75–1.15) | 1.00 (-) | 1.55* (1.02–2.35) | 1.10 (0.93–1.30) | 1.00 (-) | 0.69* (0.50–0.93) |
| Former Smoker | 1.04 (0.73–1.48) | 2.53 (0.32–19.94) | 2.60** (1.41–4.80) | 1.20 (0.92–1.58) | 1.46 (0.28–7.50) | 2.10** (1.28–3.46) | 0.96 (0.78–1.19) | 1.37 (0.38–4.91) | 0.55** (0.38–0.80) |
| **Ever Alcohol Consumption** | | | | | | | | | |
| Yes | Ref | Ref | Ref | Ref | Ref | Ref | Ref | Ref | Ref |
| No | 0.84 (0.61–1.15) | 1.38 (0.19–9.96) | 1.18 (0.73–1.93) | 0.85 (0.66–1.10) | 0.60 (0.12–2.92) | 1.15 (0.77–1.70) | 1.24* (1.01–1.52) | 1.21 (0.46–3.18) | 0.82 (0.61–1.09) |
| **Teeth Cleaning Frequency** | | | | | | | | | |
| Once a day | Ref | Ref | Ref | Ref | Ref | Ref | Ref | Ref | Ref |
| Twice a day | 1.10 (0.93–1.31) | 1.75 (0.16–19.15) | 1.28 (0.63–2.60) | 1.03 (0.90–1.18) | 2.64 (0.42–16.67) | 1.54 (0.88–2.68) | 1.03 (0.90–1.18) | 0.92 (0.28–2.96) | 0.70 (0.46–1.07) |
| Infrequent | 1.43 (0.59–3.47) | 1.12 (0.08–16.13) | 1.24 (0.70–2.19) | 0.89 (0.39–2.03) | 1.37 (0.12–15.68) | 1.22 (0.76–1.97) | 0.89 (0.39–2.03) | 0.63 (0.19–2.16) | 0.83 (0.59–1.18) |

AOR: Adjusted Odds Ratio; CI: Confidence Interval

*: p<0.05; **:p<0.01; ***:p<0.001

and Bhutan), and who were current or former smokers (in Nepal) were more likely to visit a dentist in the last 6/12 months.

Overall, we found a low prevalence of teeth cleaning twice daily, similar to previous research conducted in Nepal [17]. Brushing teeth twice a day is inversely associated with the accumulation of dental plaque [33]. However, our findings suggest that most participants were at risk of developing dental plaque, which may lead to oral and periodontal diseases [38].

The majority of participants from all three countries never visited a dentist. Regular dental visits, at least once every six months, are recommended for routine cleaning and checkups [17]. However, the dental health system in most South Asian countries is not well-organized, and high costs and lack of dental insurance may be barriers to routine visits [39]. Therefore, further exploration is needed to understand why residents of these countries do not regularly visit dentists [40].

Older participants were more likely to clean their teeth twice a week and visit a dentist within the past six or twelve months [40]. This could be due to increased oral and periodontal treatment needs with age, along with more preventive information from the health system [41]. Women were more likely to adhere to oral hygiene compared to men, aligning with findings from a study by Márquez-Arrico et al. [42]. Men are disproportionately affected by periodontal diseases due to poor oral hygiene practices and reluctance to visit a dentist regularly [43]. Further exploration is needed to understand gender disparities in oral hygiene [44]. Similar to studies in various income-level countries, increased educational attainment correlated positively with oral hygiene practices, possibly due to greater awareness among educated individuals [45, 46].

Non-smokers in Bangladesh and Nepal and non-alcohol consumers in Bhutan and Nepal were more likely to clean their teeth [47]. However, current smokers and alcohol consumers in this study were less likely to follow oral hygiene practices, putting them at risk of developing periodontal diseases [48, 49]. More Bangladeshi adults cleaned their teeth once or twice a day compared to adults in the other two countries. On the other hand, a higher rate of Bhutanese adults visited a dentist within the previous six or twelve months, possibly reflecting Bhutan's higher per capita GDP compared to Bangladesh and Nepal. In Bangladesh and Nepal, less than 3% of adults sought routine oral checkups from dentists, highlighting the need to enhance awareness about the importance of regular dental visits.

The study has limitations, including its cross-sectional nature, which precludes causal inference, and a limited number of correlates in the sample. When interpreting the results of this study, it is important to acknowledge the limitations arising from the lack of detailed information about the primary data collection methods. This caveat should be considered when assessing the robustness of our findings. As this is a secondary analysis, the researchers are restricted by the data collected in the original WHO STEPS surveys. There might be important factors influencing oral health behaviours that were not included in these surveys, and thus cannot be explored in this study. For instance, the WHO STEPS datasets did not contain information on wealth status, preventing adjustment for this variable in our analysis. Due to data limitations, urban/rural differences could not be explored in Nepal. Additionally, the lack of comprehensive oral health data in other South Asian nations prevented regional comparisons. Our oral health outcomes were limited to only two main variables, as no other commonly measured oral health outcomes were consistent across the three surveys. This study utilized secondary datasets, which provide valuable insights through extensive coverage and standardized methodologies but is limited by data quality variability and the potential misalignment of variables with our specific research objectives. The original data collection may have been impacted by factors such as participant recall bias or social desirability bias. These influences could compromise the accuracy of the findings in this secondary analysis. However, the study's strengths

include the use of nationally representative data and validated questionnaires, reducing the likelihood of information bias.

This study holds significant implications for public health policy. The governments of Bangladesh, Bhutan, and Nepal, along with their regulatory bodies and health policy experts, must prioritize promoting dental healthcare. This involves emphasizing education and training to establish a robust network of trained medical doctors, dentists, and dental hygienists. Additionally, creating transdisciplinary networks and workshops among professional and research organizations is crucial for developing a comprehensive 'National Policy for Oral Healthcare.' Such a network should include dental professionals, medical authorities, media outlets, public health experts, and government healthcare bodies. According to the Global Health Observatory data released by the WHO in 2021, for every 10,000 individuals, Bangladesh has only 0.69 dentists, while Bhutan and Nepal have 0.97 and 1.36 dentists, respectively [43]. In contrast, even developed and less populated countries like Iceland and Finland have approximately eight and ten dentists per 10,000 residents, respectively [43]. Thus, it is crucial to utilize available resources to enhance oral healthcare awareness and practices in countries such as Bangladesh, Nepal, and Bhutan. Developing healthcare infrastructures in rural areas and academic dental institutions to increase the dentist ratio in these countries will require time and funding. However, employing remote healthcare technologies and teledentistry can help raise awareness among rural populations and bridge the gap due to the limited number of dentists [44, 45].

For children in the preschool period, early oral hygiene practices are essential for preventing dental caries, the most common chronic disease in childhood. Good oral hygiene habits, including regular brushing and dental check-ups, can prevent tooth decay and its associated pain and complications, affecting eating, speaking, and overall quality of life [50, 51]. Moreover, early dental visits help inculcate a positive attitude towards dental health, laying the foundation for healthy oral hygiene practices that can last a lifetime [52]. Previous studies also showed that poor knowledge and awareness of parents and caregivers about potential risk factors of oral health are associated with infant and child oral health risk status [51, 53]. There is a need for school-based oral health prevention programs, emphasizing the blend of theoretical knowledge and practical application for effective learning [54, 55]. Educating mothers, including pregnant women, about the importance of oral care for their children can yield long-term benefits. There is a need for more studies in these countries to address the gap in knowledge and the limited availability and accessibility of public dental services [46].

Campaigns and promotional activities targeting school-going children at the community level should be conducted to educate them about the harmful effects of smoking, tobacco, and alcohol consumption. Social media platforms, podcasts, television commercials, and print media, such as newspapers, leaflets, and magazines, may serve as valuable tools for these oral health promotional activities [48, 49]. Additionally, implementing upstream approaches like sugar taxation could play a pivotal role in improving the population's dental health [56]. Future studies should also focus on oral health, the prevalence of periodontal diseases, and their correlation with systemic diseases in these three nations.

## Conclusions

This study reported the prevalence of oral hygiene practices (specifically teeth cleaning frequency) and dental service utilization, along with associated socio-demographic factors, in three South Asian countries: Bangladesh, Bhutan, and Nepal. The prevalence of cleaning teeth once a day was approximately 90% in all three countries. However, the frequency of cleaning teeth twice daily and visiting a dentist was notably lower. The study identified age, gender, and educational attainment as significant factors associated with oral hygiene practices across these

countries. To further improve oral health, it is recommended that public health initiatives in these countries focus on increasing awareness of the benefits of cleaning teeth twice daily and having regular dental check-ups. Additionally, integrating oral health education into school curricula could help instill good dental habits from a young age, potentially reducing future dental health issues.

## Supporting information

**S1 Table. Distribution of the respondents regarding the materials used to clean teeth in Bangladesh and Nepal.**
(DOCX)

**S2 Table. Distribution of the respondents regarding the reason for visiting a dentist in Bangladesh and Nepal.**
(DOCX)

**S3 Table. Crude and adjusted prevalence ratios and odds ratio for the factors associated with cleaning teeth at least once a day in Bangladesh.**
(DOCX)

**S4 Table. Crude and adjusted prevalence ratios and odds ratio for the factors associated with cleaning teeth at least once a day in Bhutan.**
(DOCX)

**S5 Table. Crude and adjusted prevalence ratios and odds ratio for the factors associated with cleaning teeth at least once a day in Nepal.**
(DOCX)

**S6 Table. Crude and adjusted prevalence ratios and odds ratio for the factors associated with cleaning teeth at least twice a day in Bangladesh.**
(DOCX)

**S7 Table. Crude and adjusted prevalence ratios and odds ratio for the factors associated with cleaning teeth at least twice a day in Bhutan.**
(DOCX)

**S8 Table. Crude and adjusted prevalence ratios and odds ratio for the factors associated with cleaning teeth at least twice a day in Nepal.**
(DOCX)

**S9 Table. Crude and adjusted prevalence ratios and odds ratio for the factors associated with visiting a dentist in last six months in Bangladesh.**
(DOCX)

**S10 Table. Crude and adjusted prevalence ratios and odds ratio for the factors associated with visiting a dentist in last six months in Bhutan.**
(DOCX)

**S11 Table. Crude and adjusted prevalence ratios and odds ratio for the factors associated with visiting a dentist in last six months in Nepal.**
(DOCX)

**S12 Table. Crude and adjusted prevalence ratios and odds ratio for the factors associated with visiting a dentist in last twelve months in Bangladesh.**
(DOCX)

**S13 Table. Crude and adjusted prevalence ratios and odds ratio for the factors associated with visiting a dentist in last twelve months in Bhutan.**
(DOCX)

**S14 Table. Crude and adjusted prevalence ratios and odds ratio for the factors associated with visiting a dentist in last twelve months in Nepal.**
(DOCX)

**S15 Table. Crude and adjusted prevalence ratios and odds ratio for the factors associated with never visiting a dentist in Bangladesh.**
(DOCX)

**S16 Table. Crude and adjusted prevalence ratios and odds ratio for the factors associated with never visiting a dentist in Bhutan.**
(DOCX)

**S17 Table. Crude and adjusted prevalence ratios and odds ratio for the factors associated with never visiting a dentist in Nepal.**
(DOCX)

## Author Contributions

**Conceptualization:** Rajat Das Gupta, Shams Shabab Haider, Shah Saif Jahan, Md. Irteja Islam, Ananna Mazumder, Muhammad Sohail Zafar, Nazeeba Siddika, Ehsanul Hoque Apu.

**Data curation:** Rajat Das Gupta, Shams Shabab Haider, Shah Saif Jahan, Md. Irteja Islam, Ehsanul Hoque Apu.

**Formal analysis:** Rajat Das Gupta, Muhammad Sohail Zafar, Ehsanul Hoque Apu.

**Funding acquisition:** Rajat Das Gupta.

**Investigation:** Rajat Das Gupta, Shah Saif Jahan, Md. Irteja Islam, Ananna Mazumder, Muhammad Sohail Zafar, Nazeeba Siddika.

**Methodology:** Rajat Das Gupta, Shams Shabab Haider, Shah Saif Jahan, Md. Irteja Islam, Ananna Mazumder, Muhammad Sohail Zafar, Nazeeba Siddika, Ehsanul Hoque Apu.

**Project administration:** Rajat Das Gupta, Shams Shabab Haider, Shah Saif Jahan.

**Resources:** Rajat Das Gupta, Md. Irteja Islam.

**Software:** Rajat Das Gupta.

**Supervision:** Rajat Das Gupta, Shams Shabab Haider, Muhammad Sohail Zafar, Ehsanul Hoque Apu.

**Validation:** Rajat Das Gupta.

**Visualization:** Rajat Das Gupta.

**Writing – original draft:** Rajat Das Gupta, Shams Shabab Haider, Shah Saif Jahan, Md. Irteja Islam, Ananna Mazumder, Muhammad Sohail Zafar, Nazeeba Siddika, Ehsanul Hoque Apu.

**Writing – review & editing:** Rajat Das Gupta, Ehsanul Hoque Apu.

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
