## [Decision Letter · Decision Letter 0]

9 Feb 2024

PGPH-D-24-00051

Dental service utilization and oral hygiene practice in Bangladesh, Bhutan, and Nepal: Findings from nationally representative surveys

Dear Dr. Gupta,

Thank you for submitting your manuscript to PLOS Global Public Health. After careful consideration, we feel that it has merit but does not fully meet PLOS Global Public Health’s publication criteria as it currently stands. Therefore, we invite you to submit a revised version of the manuscript that addresses the points raised during the review process.

We look forward to receiving your revised manuscript.

Kind regards,

Palash Chandra Banik, MPhil

Academic Editor

Journal Requirements:

Additional Editor Comments (if provided):

Please address all the comments from the reviewers. 

Reviewers' comments:

1. Please send a completed 'Competing Interests' statement, including any COIs declared by your co-authors. If you have no competing interests to declare, please state "The authors have declared that no competing interests exist". 

If you did not receive any funding for this study, please simply state: “The authors received no specific funding for this work.”"

Reviewer's Responses to Questions

**Comments to the Author**

1. Does this manuscript meet PLOS Global Public Health’s publication criteria? Is the manuscript technically sound, and do the data support the conclusions? The manuscript must describe methodologically and ethically rigorous research with conclusions that are appropriately drawn based on the data presented.

Reviewer #1: Yes

Reviewer #2: Yes

2. Has the statistical analysis been performed appropriately and rigorously?

Reviewer #1: Yes

Reviewer #2: Yes

3. Have the authors made all data underlying the findings in their manuscript fully available (please refer to the Data Availability Statement at the start of the manuscript PDF file)?

Reviewer #1: Yes

Reviewer #2: Yes

4. Is the manuscript presented in an intelligible fashion and written in standard English?

Reviewer #1: Yes

Reviewer #2: Yes

5. Review Comments to the Author

Reviewer #1: interesting topic.

well designed paper

Good English language

Adequate tables.

The introducion and Discussion sections should be implemented with the topic about the importance of oral hygiene from the pregnancy and in pre-school period (please cite PMID: 29178747; PMID: 28462177; PMID: 30789195; PMID: 31110611

Reviewer #2: In view that you assessed only one oral hygiene practice (toothbrushing) and one dental service utilization variable (last visit to the dentist), the title is too widely worded hence misleading.

Since the fundamental reason for toothbrushing and dental visits are dental caries and periodontitis, some numerical data re. prevalence of caries and periodontitis in the 3 countries is relevant in the introduction to put things in proper perspective.

References are not pin-pointed. For example Ref 17 is several times used with no page numbers. It is very difficult for the reader to widen his view of the issue from the reference this way. It is not feasable to read the whole refereed book, or even article which is many pages long, like ref 16.

6. PLOS authors have the option to publish the peer review history of their article (what does this mean?). If published, this will include your full peer review and any attached files.

**Do you want your identity to be public for this peer review?** For information about this choice, including consent withdrawal, please see our Privacy Policy.

Reviewer #1: No

Reviewer #2: No

---

## [Decision Letter · Decision Letter 1]

23 Apr 2024

PGPH-D-24-00051R1

Prevalence and associated factors of last dental visit and teeth cleaning frequency in Bangladesh, Bhutan, and Nepal: Findings from nationally representative surveys

Dear Dr. Gupta,

Thank you for submitting your manuscript to PLOS Global Public Health. After careful consideration, we feel that it has merit but does not fully meet PLOS Global Public Health’s publication criteria as it currently stands. Therefore, we invite you to submit a revised version of the manuscript that addresses the points raised during the review process.

Please make sure to address the comments raised by the reviewer in a point wise manner and in text.

We look forward to receiving your revised manuscript.

Kind regards,

Pragati B Hebbar

Academic Editor

Journal Requirements:

Additional Editor Comments (if provided):

Reviewers' comments:

Reviewer's Responses to Questions

**Comments to the Author**

1. If the authors have adequately addressed your comments raised in a previous round of review and you feel that this manuscript is now acceptable for publication, you may indicate that here to bypass the “Comments to the Author” section, enter your conflict of interest statement in the “Confidential to Editor” section, and submit your "Accept" recommendation.

Reviewer #2: All comments have been addressed

Reviewer #3: (No Response)

2. Does this manuscript meet PLOS Global Public Health’s publication criteria? Is the manuscript technically sound, and do the data support the conclusions? The manuscript must describe methodologically and ethically rigorous research with conclusions that are appropriately drawn based on the data presented.

Reviewer #2: Yes

Reviewer #3: Partly

3. Has the statistical analysis been performed appropriately and rigorously?

Reviewer #2: Yes

Reviewer #3: Yes

4. Have the authors made all data underlying the findings in their manuscript fully available (please refer to the Data Availability Statement at the start of the manuscript PDF file)?

Reviewer #2: Yes

Reviewer #3: Yes

5. Is the manuscript presented in an intelligible fashion and written in standard English?

Reviewer #2: Yes

Reviewer #3: Yes

6. Review Comments to the Author

Reviewer #2: No comments

Reviewer #3: Please address the comments

7. PLOS authors have the option to publish the peer review history of their article (what does this mean?). If published, this will include your full peer review and any attached files.

**Do you want your identity to be public for this peer review?** For information about this choice, including consent withdrawal, please see our Privacy Policy.

Reviewer #2: No

Reviewer #3: **Yes: **Rajeev B R

---

## [Decision Letter · Decision Letter 2]

23 May 2024

PGPH-D-24-00051R2

Prevalence and associated factors of last dental visit and teeth cleaning frequency in Bangladesh, Bhutan, and Nepal: Findings from nationally representative surveys

Dear Dr. Gupta,

Thank you for submitting your manuscript to PLOS Global Public Health. After careful consideration, we feel that it has merit but does not fully meet PLOS Global Public Health’s publication criteria as it currently stands. Therefore, we invite you to submit a revised version of the manuscript that addresses the points raised during the review process.

We look forward to receiving your revised manuscript.

Kind regards,

Pragati B Hebbar

Academic Editor

Journal Requirements:

Additional Editor Comments (if provided):

Dear Dr. Rajat Das Gupta,

Thank you for taking the time to revise the manuscript based on the points identified by the reviewer. On review of the revised submission it is noted that the manuscript still needs some more clarification. Please find detailed suggestions by the reviewer to improve the rigour of the paper and include them in the manuscript and resubmit. Look forward to receiving the revised manuscript addressing all the concerns raised.

Reviewers' comments:

Reviewer's Responses to Questions

**Comments to the Author**

1. If the authors have adequately addressed your comments raised in a previous round of review and you feel that this manuscript is now acceptable for publication, you may indicate that here to bypass the “Comments to the Author” section, enter your conflict of interest statement in the “Confidential to Editor” section, and submit your "Accept" recommendation.

Reviewer #3: (No Response)

2. Does this manuscript meet PLOS Global Public Health’s publication criteria? Is the manuscript technically sound, and do the data support the conclusions? The manuscript must describe methodologically and ethically rigorous research with conclusions that are appropriately drawn based on the data presented.

Reviewer #3: Partly

3. Has the statistical analysis been performed appropriately and rigorously?

Reviewer #3: Yes

4. Have the authors made all data underlying the findings in their manuscript fully available (please refer to the Data Availability Statement at the start of the manuscript PDF file)?

Reviewer #3: No

5. Is the manuscript presented in an intelligible fashion and written in standard English?

Reviewer #3: Yes

6. Review Comments to the Author

Reviewer #3: The manuscript needs some more clarification. Authors could consider the suggestions and include them in the manuscript to proceed further

7. PLOS authors have the option to publish the peer review history of their article (what does this mean?). If published, this will include your full peer review and any attached files.

**Do you want your identity to be public for this peer review?** For information about this choice, including consent withdrawal, please see our Privacy Policy.

Reviewer #3: No

---

## [Editor Report · Decision Letter 3]

2 Jul 2024

Prevalence and associated factors of last dental visit and teeth cleaning frequency in Bangladesh, Bhutan, and Nepal: Findings from nationally representative surveys

PGPH-D-24-00051R3

Dear Dr. Gupta,

We are pleased to inform you that your manuscript 'Prevalence and associated factors of last dental visit and teeth cleaning frequency in Bangladesh, Bhutan, and Nepal: Findings from nationally representative surveys' has been provisionally accepted for publication in PLOS Global Public Health.

Best regards,

Pragati B Hebbar

Academic Editor

Thank you for the point wise response and addressing all the comments raised by the reviewers satisfactorily.